# The Effects of Multicomponent Training on Clinical, Functional, and Psychological Outcomes in Cardiovascular Disease: A Narrative Review

**DOI:** 10.3390/medicina61050822

**Published:** 2025-04-29

**Authors:** Luca Poli, Alessandro Petrelli, Francesco Fischetti, Stefania Morsanuto, Livica Talaba, Stefania Cataldi, Gianpiero Greco

**Affiliations:** 1Department of Translational Biomedicine and Neuroscience (DiBraiN), University of Study of Bari, 70124 Bari, Italy; luca.poli@uniba.it (L.P.); alessandro.petrelli@uniba.it (A.P.); gianpiero.greco@uniba.it (G.G.); 2Department of Education and Sport Sciences, Pegaso Telematic University, 80143 Naples, Italy; stefania.morsanuto@unipegaso.it (S.M.); stefania.cataldi@unipegaso.it (S.C.); 3Department of Surgical Pathology, University of Pisa, 56126 Pisa, Italy; livicatalaba@gmail.com

**Keywords:** physical activity, aerobic training, resistance training, balance, flexibility, blood pressure, hypertension, quality of life

## Abstract

Cardiovascular diseases (CVDs) remain the leading cause of death globally. In recent years, interest in multicomponent interventions has grown as a response to the multifactorial complexity of CVDs. However, the literature still shows little systematic investigation into the effectiveness of multicomponent training (MCT) in the field of CVDs, accompanied by terminological confusion. This study aims to summarize and critically appraise the recent literature through a narrative review. A narrative review was conducted, synthesizing evidence from studies published between 2010 and January 2025. The databases searched included PubMed, Scopus, and Google Scholar using predefined search terms related to CVDs and MCT, and medical subject headings (MeSHs) and Boolean syntax. Two team authors independently extracted relevant information from the included studies. MCT significantly improved hemodynamic parameters in CVD patients, with reductions in systolic, diastolic, mean blood pressure, and heart rate. Physical fitness measures showed consistent enhancements whereas anthropometric improvements often corresponded with blood pressure reductions. Psychological outcomes varied across studies, with intervention duration emerging as a key factor in effectiveness. MCT interventions could lead to improvements in clinical outcomes, risk factor reduction, and patient adherence. Although findings on psychological parameters remain inconsistent, the overall evidence supports their integration into both clinical and community settings.

## 1. Introduction

The global burden of cardiovascular pathologies continues to escalate, presenting significant challenges to healthcare systems worldwide. These conditions substantially affect millions of individuals, diminishing both longevity and quality of life (QoL) [1,2]. The intricate pathophysiology of cardiovascular disease (CVD) necessitates comprehensive management strategies that address both somatic and psychological dimensions [3,4]. Multiple risk determinants influence cardiovascular health, including excessive adiposity, metabolic dysregulation, advanced age, and insufficient physical activity (PA) [5].

Among various interventions, structured (PA) protocols have emerged as pivotal in both preventive and therapeutic cardiovascular care. Evidence suggests that regular exercise enhances cardiac performance, augments functional capacity, and contributes significantly to overall well-being beyond mere weight control [6,7,8,9].

The spectrum of cardiovascular pathologies encompasses numerous distinct clinical aspects affecting the heart and more broadly the cardiovascular system. Optimal management typically requires an integrated approach combining pharmacological therapy with lifestyle optimization. Structured pre- and post-cardiac rehabilitation initiatives, incorporating supervised exercise sessions, nutritional guidance, and psychological support, play a fundamental role in reducing recurrence risk and enhancing prognosis [10,11,12].

Traditionally, aerobic training has dominated exercise recommendations for cardiovascular patients. Evidence indicates that aerobic protocols enhance exercise tolerance, optimize cardiac output, reduce resting heart rate, and improve vascular reactivity [13,14]. Additionally, these regimens positively influence lipid metabolism, glucose homeostasis, and body composition, collectively addressing multiple cardiovascular risk factors [15,16]. More recently, resistance training has gained recognition as an essential component of cardiovascular rehabilitation.

Research indicates that progressive resistance training contributes to enhanced muscular performance, augmented lean tissue mass, and improved insulin sensitivity [17]. These adaptations collectively reduce frailty risk and support cardiovascular function. Furthermore, appropriately prescribed resistance exercises can favorably modulate hemodynamic parameters and endothelial function [18,19]. The physiological mechanisms underlying the cardiovascular benefits of resistance training are multifaceted. Enhanced muscle mass promotes improved glucose utilization and insulin action, which is particularly relevant for patients with metabolic syndrome or type 2 diabetes, conditions frequently accompanying cardiovascular disorders. In addition to these metabolic effects, resistance protocols have been demonstrated to be effective in reducing psychological stress and anxiety, which are recognized contributors to cardiovascular risk [20].

Cardiovascular impairments often compromise circulatory efficiency, potentially affecting musculotendinous structures and connective tissue integrity. Inadequate tissue perfusion may promote fibrotic changes, altered collagen cross-linking, and increased muscle stiffness, ultimately restricting range of motion and functional capacity [21]. Flexibility-focused interventions, incorporating stretching and mobility exercises, can enhance joint function and tissue elasticity, potentially improving functional outcomes in cardiovascular patients.

Emerging evidence suggests that systematic stretching protocols may reduce arterial stiffness, enhance endothelial function, and favorably modulate autonomic balance [22]. The mechanisms underlying these benefits are still not fully understood; recently, even the role of the so-called extracellular vesicles (EVs) and their cargo was explored, suggesting that different exercise modalities can differently modulate this EVs [23].

Given the complexity of cardiovascular pathophysiology, researchers and clinicians increasingly recognize the potential benefits of multicomponent training (MCT) programs, which systematically integrate aerobic, resistance, balance, and/or flexibility exercises [24]. This integrated approach addresses multiple fitness domains simultaneously, enhancing cardiovascular endurance, muscular strength, joint mobility, and postural stability within a unified framework [25,26]. A notable advantage of MCT protocols is their comprehensive nature within time-efficient parameters, addressing multiple physiological systems within single sessions. This characteristic may promote adherence in cardiovascular patients, who frequently show suboptimal compliance with extended or monotonous exercise regimens [27,28]. MCT could be particularly effective in improving both peripheral and central hemodynamic parameters, cardiorespiratory fitness, muscular function, and body composition in different populations, including patients with CVDs [25,29,30,31,32,33,34]. This training approach has demonstrated favorable effects on both physiological parameters and biochemical markers, including inflammatory mediators and lipid profiles [35].

Despite this evidence, uncertainty remains about the actual effectiveness of this type of training, especially in the context of CVDs. This is compounded by a terminological issue that often leads to the assimilation of different terms, such as combined training or multicomponent protocols, which do not fully reflect the meaning of MCT. Furthermore, uncertainty persists regarding whether the observed benefits of MCT derive from the additive effects of different exercise modalities or simply from increased overall exercise volume. To date, no review has analyzed the psychophysical effects of properly defined MCT in subjects with CVDs. Clarifying these effects and highlighting areas of agreement or inconsistency is crucial in terms of informing clinical exercise prescription.

Therefore, this review aims to critically evaluate the current evidence regarding MCT effects on physiological and psychological parameters in cardiovascular patients. By synthesizing available data, this review seeks to clarify the efficacy of the MCT approach and identify future research priorities to optimize exercise prescription for this growing population.

## 2. Materials and Methods

This narrative review was carried out following the narrative review checklist [36]. Comprehensive research on Medline In-Process and other Non-Indexed Citations was conducted (prior to 28 February 2025), using MEDLINE (PubMed), Web of Science (WoS), and Google Scholar to retrieve relevant articles. The literature search was performed via medical subject headings (MeSHs) and Boolean syntax. Controlled terms were used to search for studies (“multicomponent training” OR “multicomponent exercise” OR “combined training” OR “multiple component exercise” OR “multimodal exercise”) AND (“cardiovascular disease” OR “CVD” OR “heart disease” OR “coronary artery disease” OR “myocardial infarction” OR “heart failure” OR “hypertension” OR “stroke” OR “peripheral artery disease”) AND (“prevention” OR “treatment” OR “management” OR “outcomes” OR “mortality” OR “morbidity”). The following filters were used: full text, human studies, and English language. After candidate articles were collected, further identification was conducted based on the inclusion and exclusion criteria.

### Study Selection

The inclusion criteria were only English-language original peer-reviewed articles, randomized and non-randomized studies, and observational and pilot studies published from January 2010 to January 2025. The excluded records were review articles, meta-analyses, practical guidelines, unpublished studies, editorials, letters to the editor, and essays, although they were used as an added measure to ensure search comprehensiveness and were used as references. Furthermore, this review included studies that had a control and/or comparison group, except for those involving participants with severe conditions that did not allow safe exercise practices (e.g., severe pulmonary hypertension, uncontrolled diabetes, unstable coronary artery diseases, or kidney failure). The search was only restricted to the type of exercise and not to the frequency, intensity, or time (FITT) of exercise, or to gender, age, sample size, or length of follow-up. Since MCT is defined as a training protocol that incorporates at least three exercise modalities (such as aerobic training, strength/resistance training, flexibility, balance, and coordination) [24], all studies that labeled their intervention as multicomponent but included fewer than three exercise modalities were excluded. Additionally, studies that used the term “multicomponent” to describe the integration of physical activity/exercise with a nutritional or broader lifestyle approach were also excluded. Finally, this review included only studies with supervised training sessions. Two team authors independently extracted relevant information from the included studies: author, year of publication, study design, number and age of participants, health condition, workload, treatment, duration of intervention, and main outcomes. Any disagreements were resolved by consensus. The characteristics of the studies were summarized, and data on the effects of MCT on physiological and psychological variables were synthesized.

## 3. Results

### 3.1. Identification of Studies

At the end of the selection process, 2087 articles were extracted, of which n = 726 were from Web of Science, n = 318 were from PubMed, and n = 1043 were from Google Scholar. Each title and abstract were screened for relevance, removing review articles, unpublished studies, meta-analyses, practical guidelines, editorials, letters to the editor, and essays (n = 1865). Thereafter, the search strategy was based on the assessment of the full text of the remaining 139 articles to verify their eligibility. Lastly, 10 research articles specifically focusing on the psychophysiological effects of MCT in subjects with CVDs were included (Figure 1), and may be divided by subheadings. This should provide a concise and precise description of the experimental results, their interpretation, and the experimental conclusions that can be drawn.

### 3.2. Study Characteristics

This narrative review provides an evaluation of 10 studies [33,37,38,39,40,41,42,43,44,45] inquiring in-depth into the psychophysiologically induced changes to MCT in subjects with CVDs. Most of them (n = 7) [37,39,40,41,43,44,45] pertain to subjects with hypertension; whereas, one work (n = 1) [38] focuses on cardiovascular risk factors, one (n = 1) on post-stroke subjects with frailty [42], and just one study (n = 1) focuses on subjects with different CVDs [33]. Table 1 summarizes the characteristics of the studies included.

## 4. Discussion

### 4.1. Effects of Multicomponent Training on Hemodynamic Parameters

PA and exercise can induce both acute and chronic hemodynamic adaptations. During exercise, cardiac output increases substantially due to elevated heart rate and stroke volume, while systolic blood pressure (SBP) rises proportionally to intensity. Blood flow redistributes dramatically, with active muscles receiving up to 85% of cardiac output. Regular exercise leads to beneficial chronic adaptations including reduced resting heart rate, increased stroke volume, lowered blood pressure, improved vascular compliance, and enhanced endothelial function. Different exercise modalities produce varying hemodynamic responses: aerobic exercise creates volume loading leading to eccentric cardiac hypertrophy; resistance training produces pressure loading potentially causing concentric hypertrophy; and isometric exercise induces dramatic blood pressure increases with minimal cardiac output changes [46,47,48]. These hemodynamic adaptations represent key mechanisms through which exercise could reduce CVD risk.

The combination of multiple exercise modalities within a single training protocol for each session, initially considered a potential cause of interference between the adaptations induced by different types of exercise (i.e., aerobic and resistance training), was later reassessed as a time-efficient approach when properly calibrated [49,50,51]. In recent years, it has therefore gained increasing popularity as a training strategy for both healthy individuals and those with medical conditions, including CVDs. Nonetheless, there appears to be some terminological confusion regarding these more recent training modalities. Terms such as combined, concurrent, multimodal, and MCT are often used interchangeably, despite each referring to a distinct training approach. Specifically, MCT denotes, as previously stated, the combination of at least three different types of exercise, such as aerobic training, strength/resistance training, flexibility, balance, and coordination [24].

Focusing on that last approach, this review found that Coelho-Júnior, Asano et al. [40] reported significant blood pressure reductions after 6 months of multicomponent training (MCT) in both normotensive and hypertensive older adults. The most notable improvements were seen in uncontrolled hypertensive individuals, with marked decreases in systolic (−8 mmHg), diastolic (−11.1 mmHg), mean blood pressure, heart rate (−6.8 bpm), and double product. These changes were accompanied by improvements in body composition. The intervention consisted of twice-weekly, 48 min sessions over 26 weeks, involving moderate-intensity functional, resistance, balance, and walking exercises. While the findings support MCT’s efficacy, especially in patients with uncontrolled hypertension, the quasi-experimental design, lack of a control group, and absence of objective dietary monitoring may limit the validity of the results due to potential confounding factors.

Consistently, Moraes et al. [43] investigated the effects of two weekly MCT sessions on blood pressure in older hypertensive patients, reporting a modest reduction in systolic (−3.2 mmHg) and diastolic (−2.0 mmHg) blood pressure after 12 weeks. However, the small sample size (n = 36) and the non-blinded blood pressure assessment could have biased the results. It should be noted that the reduction in blood pressure was accompanied by a BMI reduction, as observed in a previous study [40], that could have contributed to the blood pressure reduction. Additionally, it was observed that individuals with higher baseline blood pressure experienced greater reductions, but this hypothesis was not formally tested because of the limited sample size.

In contrast, a subsequent study by the same author [41] showed no significant improvements in hemodynamic parameters after six months of similar MCT in hypertensive patients with osteoarthritis. The lack of an effect on blood pressure could be attributed to a population with hypertension already controlled pharmacologically or insufficient training intensity to induce cardiovascular adaptations. Similarly, no significant change in heart rate was found. This difference could be explained by the type of exercise included in the MCT, while the previous study [40] incorporated a more pronounced aerobic component, this study emphasized balance and mobility exercises, which were less cardiovascular demanding.

Conversely, the study by Baptista et al. [37] analyzed an MCT intervention in combination with different pharmacological treatments for hypertension (diuretics, calcium channel blockers, and β-blockers) over a two-year period. All groups showed a reduction in SBP, suggesting that exercise has a positive effect independent of drug treatment. This result contrasts with the study by Coelho-Júnior, Gonçalvez et al. [41], strengthening the hypothesis that a longer period of exercise may be necessary to achieve significant hemodynamic effects.

The reduction in blood pressure was accompanied by an improvement in the lipid profile and body composition, with a decrease in waist circumference in the diuretic- and β-blocker-treated groups. This suggests that the association between exercise and drug treatment may amplify cardiovascular benefits, an aspect not evaluated in previous studies [40,41]. Furthermore, the longer duration of the intervention may explain why only Baptista et al. [37] reported significant effects on blood pressure under all conditions.

In a recent randomized controlled study [33] with the aim of comparing the effects of MCT and aerobic exercise (AT) only on hemodynamic parameters, physical fitness, and QoL in sedentary elderly with CVDs, the authors observed that both exercise protocols had similar effects on the hemodynamic parameters (blood pressure and resting heart rate). This study is particularly relevant because, unlike the others analyzed, it seeks to clarify whether an integrated and combined approach (MCT) is as effective as aerobic training only in improving parameters related to cardiovascular health.

However, several limitations must be considered. First, most of the recruited population had hypertension and exhibited heterogeneous CVDs (hypertension, valvular heart disease, aortic valve disease, atrial fibrillation, and previous myocardial infarction), unlike previous studies [37,40,41] where subjects had only hypertension. The sample used, although divided into three groups (MCT, AT, and CG) was small, with 33 subjects compared with 183 [40], 99 [41], and 96 [37]. Finally, the short duration of the intervention, 10 weeks compared to the 3 months [43], 6 months [40,41], or 2 years [37] of the previous studies, could represent a caveat.

It should be noted that, similar to Baptista et al. [37], the subjects were also under drug treatment, suggesting that it may be an effective adjunct to standard treatment and, contrary to Coelho-Júnior, Asano et al. [40] and Moraes et al. [43], the improvement in hemodynamic parameters was not paired with a BMI reduction, suggesting that these changes may be independent of changes in body weight.

### 4.2. Effects of Multicomponent Training on Physical Fitness Parameters

MCT has demonstrated beneficial effects on various parameters of physical fitness and anthropometric measures in populations with CVDs. Regarding the anthropometric parameters, significant reductions in body weight, body mass index (BMI), and waist circumference have been observed [37,45], although some studies have reported no statistically significant changes [41]. This variability may be attributable to differences in intervention duration and intensity, as well as specific population characteristics.

For instance, Baptista et al. [37] observed substantial improvements following a 24-month intervention, whereas Trapé et al. [45] reported reductions in weight and BMI after just 12 weeks in hypertensive patients. This temporal discrepancy suggests that, while appreciable changes can emerge in the short term, more pronounced and sustainable effects could be observed upon extended intervention durations. Notably, the efficacy of MCT in modifying body composition appears greater in individuals with less controlled blood pressure [40], indicating a possible modulatory role of hemodynamic status in the responsiveness to exercise.

In terms of physical functionality, a generalized pattern of improvement emerges, albeit with variable effect sizes. In hypertensive individuals, regardless of the level of blood pressure control, significant enhancements in balance and gait performance have been noted [39,41]. Consistent gains in the muscular strength of both the upper and lower limbs have also been observed, as evidenced by Baptista et al. [37] and further confirmed by Poli et al. [33], who also demonstrated the superiority of MCT over isolated aerobic training in strengthening the lower limbs and dominant handgrip strength.

Pepera et al. [44] reported significant improvements in balance and mobility after only 8 weeks of training, suggesting that certain functional parameters may exhibit a faster response compared to anthropometric outcomes. Particularly noteworthy is the impact of MCT on frailty reduction and improvements in activities of daily living. Similarly, Luo et al. [42] reported significant decreases in frailty scores alongside increases in functional independence following a 12-week intervention.

The study by Baptista et al. [37] demonstrated that a 24-month MCT intervention led to significant functional improvements in hypertensive patients, regardless of the medication used, highlighting the value of prolonged programs. The study also emphasized enhanced self-efficacy, a key factor in long-term exercise adherence. In contrast, variability across other studies may stem from differences in training frequency, intensity, duration, and exercise type. Moreover, most studies had limited follow-up, which hinders evaluation of the long-term effects and adds methodological limitations.

It must be noted that, in a recent study [33], when comparing MCT with aerobic training only in elderly cardiovascular patients, significant improvements were observed in both groups without substantial differences between modalities, except for lower limb strength and dominant handgrip strength. However, although both the aerobic and MCT protocol improve physical fitness parameters, MCT seems to address critical physical deficits, especially lower limb weakness, that could improve independence, reduce fall risk, and enhance long-term health outcomes. MCT should be considered a comprehensive strategy for managing CVDs, especially in elderly populations.

### 4.3. Effects of Multicomponent Training on Psychological Parameters

In recent years, MCT has gained increasing attention in the fields of health and psychological well-being. Numerous studies have highlighted the benefits of this type of training on QoL, stress reduction, and mood enhancement [26,52,53]. However, while the scientific literature has investigated the effects of MCT in healthy populations or those at risk of cognitive decline, there is a relative lack of research examining its impact on individuals with CVDs.

The study by Baptista et al. [37] showed a lack of statistical significance in the improvements in the physical components of health-related quality of life (HRQoL), measured with the short form 36 (SF-36), within the analyzed groups. This limitation was primarily attributed to the small sample sizes within each group, which might have compromised the statistical power of the analysis. Despite this methodological limitation, it is important to highlight that both the group treated with beta-blockers (βBs) and the one treated with thiazides (TDs) showed clinically relevant improvements in the physical subscales of HRQoL, suggesting a positive effect from the exercise intervention. The group treated with calcium-channel blockers (CCBs) showed more encouraging results, with significant improvements in physical functioning, bodily pain, and the physical component summary (PCS) of SF-36. An interesting observation concerns the increase in the bodily pain subscale score in the CCBs group after the intervention, which the authors interpreted as a possible effect of the “normal” adjustment to the training load, manifested as muscle soreness. This phenomenon reflects the more “active” nature of exercise-based therapy, which requires more intense and prolonged participation by patients, as well as significant behavioral changes. Such changes, although initially more difficult to achieve for some individuals, have proven particularly encouraging in improving symptoms and overall QoL.

Conversely, Luo et al. [42], reported more favorable outcomes, showing that MCT can significantly enhance QoL in individuals with CVDs. Using the SF-36, the authors demonstrated improvements across physical, psychological, biochemical, and social domains. However, the study’s short duration (12 weeks), lack of follow-up, and limited control for psychological confounders may restrict conclusions about the long-term and independent effects of the intervention.

In the most recent study [33], significant improvements in physical performance were not accompanied by changes in quality of life (QoL) or perceived health status. This may be due to the short intervention duration (10 weeks), as longer programs have shown better QoL outcomes [54], and to high baseline scores, suggesting a ceiling effect. Nonetheless, high enjoyment levels and strong adherence were reported in both the MCT and aerobic groups, indicating that MCT can be equally engaging. These findings underscore the complex and delayed relationship between physical fitness improvements and subjective well-being in CVD patients.

While the study by Luo et al. [42] provides solid evidence in favor of the psychological benefits of a multicomponent exercise intervention, the studies by Baptista et al. [37] and Poli et al. [33] show more heterogeneous results, characterized by variable or non-significant improvements in QoL and perceived well-being. This variability could be attributed to various factors, including methodological differences (sample sizes, intervention durations, and assessment tools), participant characteristics (disease severity, pharmacological regimen, and initial levels of QoL), and types of exercise intervention (modality, intensity, frequency, and duration). A common element that emerged from the studies by Baptista et al. [37] and Poli et al. [33] is the importance of adherence and exercise enjoyment as crucial factors for the success of interventions based on physical activity.

Designing training programs that prioritize participant enjoyment could represent a key strategy for improving long-term adherence, particularly in populations with CVDs who might face additional barriers to exercise.

## 5. Limitations and Further Directions

To the best of our knowledge, this is the first review to analyze the effects of MCT protocols, as effectively defined [24], on psychophysical aspects in subjects with CVDs. However, while this review provides a comprehensive overview of multicomponent interventions for CVD patients, several limitations should be acknowledged. First, despite efforts to include a broad range of studies, the review was inherently limited by the quality and heterogeneity of the available research.

The included studies varied widely in their design, intervention components, follow-up duration, outcome measures, and population characteristics. This variability makes direct comparisons challenging and limits the ability to draw firm, generalizable conclusions. Second, many of the reviewed studies lacked detailed reporting on adherence rates, and long-term sustainability, which are critical factors in determining real-world effectiveness. Without this information, it is difficult to assess how consistently interventions were delivered and whether the observed outcomes were maintained over time.

Additionally, some studies relied heavily on self-reported data for lifestyle behaviors, introducing the potential for recall bias and social desirability effects. From a methodological standpoint, many studies did not use a randomized controlled design or failed to adequately account for confounding variables.

Given these limitations, future research should prioritize conducting high-quality, adequately powered randomized controlled trials with standardized outcome measures, evaluating intervention and long-term adherence through objective measures and mixed-method approaches, investigating the mechanisms through which multicomponent interventions exert their effects, possibly through mediation and moderation analyses. By addressing these gaps, future studies can better inform evidence-based guidelines and support the scalable implementation of multicomponent strategies in the prevention and management of CVDs.

## 6. Conclusions

Multicomponent interventions, integrating different exercise modalities, are more effective than single-component strategies in preventing and managing CVDs. They improve clinical outcomes, reduce risk factors, and enhance patient adherence across various populations and settings. Evidence about modifications of QoL and perceived well-being is still conflicting. Despite the variability among studies, the overall evidence supports their use in clinical and community contexts. Future efforts should focus on improving study quality, ensuring inclusiveness, and facilitating real-world implementation to fully harness their potential in reducing the global burden of CVDs.

## Figures and Tables

**Figure 1 medicina-61-00822-f001:**
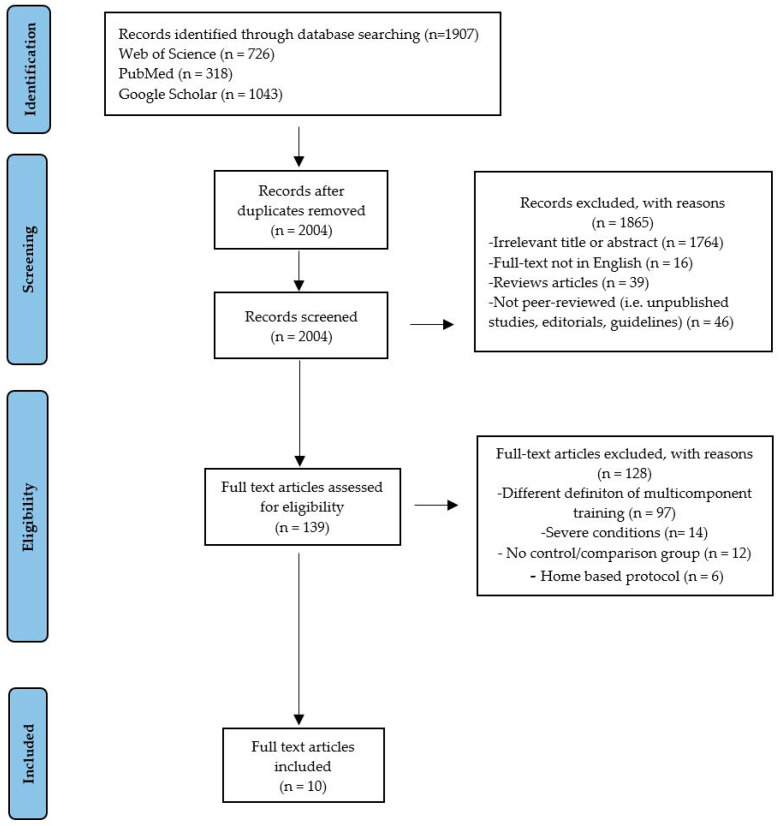
Study selection and eligibility screening flow diagram.

**Table 1 medicina-61-00822-t001:** A summary of the characteristics of the reviewed studies.

**Authors**	**Country**	**Study Design**	**Sample**	**Subjects Age (Years)**	**Health Condition**	**Workload**	**Treatment**	**Duration Intervention**	**Main Outcomes**
Moraes et al., 2012 [43]	Brazil	Quasi-experimental non- controlled study	n = 36	69.3	AH	60’ × 2/wk Subjective Effort Perception Scale, maintaining an intensity ranging between 3 and 5	Participants were using daily routine medications (ACEIs, beta blockers, diuretics, calcium-channel blockers, statins, aspirin, and oral hypoglycemic drugs)	12 weeks	EG = ↓BMI EG = ↓BG EG = ↓BP EG = ↑CST EG = ↑EFT EG = ↑SGT EG = ↑UST EG = ↓PST EG = ↓SSMT EG = No significant effects SRT
Coelho Junior et al., 2017 [39]	Brazil	Quasi-Experimental	n = 218 NTS = 101 HTS = 117	65.2 (±6.84)	HTN	48’ × 2 s/wk Subjective Effort Perception Scale, maintaining an intensity ranging between 3 and 5	Not specified	26 weeks	NTS = ↑OLST; ↑UMWS HTS = ↑OLST; ↑UMWS NTS and HTS = No significant effects STS; TUG; TUG cognitive test
Baptista et al., 2018 [37]	Portugal	Quasi-Experimental	n = 96 TDs n = 33 CCBs n = 23 βBs n = 40	67.4 (±8.7)	HTN	60’ × 3 s/wk Intensity perceived exertion scale	Individualized daily monotherapy (Indapamide 2.5 mg for TDs, Amlodipine 5 mg for CCBs, or β-blockers like Bisoprolol 5 mg, Nebivolol 5 mg, or Carvedilol 25 mg for βBs), with doses adjusted by their primary care physician to manage blood pressure and comorbidities	24 month	TDs = ↑UBS; ↑LBS; AE aerobic endurance; ↓SBP CCBs = ↑ UBS; ↑LBS; AE aerobic endurance; ↓SBP βBs = ↑ UBS; ↑LBS; AE; ↓SBP CCBs = ↓TC TDs and βBs = ↑ BM; ↑WC CCBs = ↑SF-36
Coelho-Júnior, Asano et al., 2018 [40]	Brazil	Quasi-Experimental	n = 183 Normotensives = 97 CNS = 53 UNS = 44 HTN = 86 CHS = 43 UHS = 43	65.8 (±5.2)	HTN	48’ × 2 s/wk Subjective Effort Perception Scale, maintaining an intensity ranging between 3 and 5	Not specified	26 weeks	CNS = ↓ SBP; ↓DBP UNS = ↓ SBP; ↓DBP;↓MAP UHS = ↓ SBP;↓DBP; ↓MAP UHS = ↓BMI; ↓WC UNS = ↓BMI; ↓WC CHS = no significant effects on anthropometric parameters
Coelho-Júnior, Gonçalvez et al., 2018 [41]	Brazil	Quasi-Experimental	n = 99 NTS-OA = 44 HTS-OA = 55	66.2 (±5.4)	HTN + lower limb OA	48’ × 2 s/wk Subjective Effort Perception Scale, maintaining an intensity ranging between 3 and 5	Participants used only antihypertensive medications (HTS-OA group). Some used analgesics, anti-inflammatory drugs, and/or muscle relaxants occasionally (every 15–30 days).	26 weeks	NTS-OA = no significant effects on anthropometric parameters HTS-OA = no significant effects on anthropometric parameters NTS-OA = ↑ OLST HTS-OA = ↑OLST NTS-OA = ↑UMWS HTS-OA = ↑UMWS HTS-OA = no significant effects STS; TUG; TUG cognitive test NTS-OA = no significant effects STS; TUG; TUG cognitive test HTS-OA = ↑ HR
Pepera et al., 2021 [44]	Greece	Multicenter randomized controlled clinical trial	n = 40 EG = 20 CG = 20	79.45 (±6.52)	HTN	50’2 s/wk Intensity not specified	Not specified	8 weeks	EG = ↑SBP EG = no significant effects DBP; HR EG = ↑TUG EG = ↑BBS CG = no significant effects on SBP; DBP; HR CG = ↓TUG CG = ↓BBS
Trapé et al., 2021 [45]	Brazil	Quasi-Experimental	n = 52 HS = 26 PRE-HS = 26	61.95 (±2.86)	HTN	90’ 2 s/wk Subjective Effort Perception Scale, maintaining an intensity ranging between 13 (moderate) and 15 (intense).	Not specified	12 weeks	HS= ↑BM HS = ↑BMI HS = ↓BP HS = ↑6MWT HS = ↑STS HS = ↑EFT PREHS = ↑BM PREHS = ↑BMI PREHS = ↓BP PREHS = ↑6MWT PREHS = ↑STS PREHS = ↑EFT
Cavalcante et al., 2023 [38]	Portugal	Quasi-Experimental	n = 11 EG = 11	65.8 (±8.6)	CRFs	60’ 2 s/wk Subjective Effort Perception Scale, maintaining an intensity ranging between 11 (moderate) and 14 (intense)	Not specified	18 weeks	EG = ↑EPCs EG = No significant effects on CECs
Luo et al., 2024 [42]	China	Single-blind, randomized controlled trial	n = 125 EG = 63 CG = 62	73.26 (±6.89)	Post-stroke with frailty	75’ 1–6 s/wk Subjective Effort Perception Scale, maintaining an intensity ranging between 13 (moderate) and 15 (intense)	Not specified	12 weeks	EG = ↓FRAIL scale EG = ↑MBI EG = ↑SF-36
Poli et al., 2024 [33]	Italy	Randomized controlled study	n = 33 MTG = 12 ATG = 12 CG = 9	69.5 (±4. 9)	Different stabilized CVDs	60’ 2 s/wk Subjective Effort Perception Scale, maintaining an intensity ranging between 13 (moderate) and 15 (intense)	Not specified	10 weeks	MTG/ATG = ↓RHR; ↓P-SBP; ↓P-DBP; ↑30CST; ↑TUG; ↑HGS; ↑2MST; ↑PACES MTG vs. ATG = ↑30CST; (D)HGS

Notes: AH: arterial hypertension; EG = experimental group; BMI = body mass index; BG = blood glucose; BP = blood pressure; ↑: increase; ↓: decrease; CST = chair stand test; EFT = elbow flexor test; SGT = stationary gait test; UST = unipedal stance test; UST = unipedal stance test; SSMT = sit stand move test; SRT = sit and reach test; NTS = normotensive group; HTS = hypertensive group; HTN = hypertension; OLST = one-leg stand test; UMWS = usual and maximal walking speed test; STS = sit to stand test; TUG = timed up and go; TDs = Thiazide-related diuretics; CCBs; calcium-channel blockers; βBs = β-blockers; SFB = senior fitness battery; SBP = systolic blood pressure; TC = total cholesterol; UBS = upper Body strength; LBS = lower body strength; BM = body mass; WC = waist circumference; SF-36 = Medical Outcomes Study 36-item Short-form Health Study questionnaire; CNS = controlled normotensives; UNS = uncontrolled normotensives; CHS = controlled hypertensives; UHS = uncontrolled hypertensives; DBP = diastolic blood pressure; MAP = mean arterial pressure; OA = lower limb osteoarthritis; HR = heart rate; BBS = Berg balance scale; CRFs = cardiovascular risk factors; EPCs = endothelial progenitor; CECs = circulating endothelial cells; MBI = Modified Barthel Index; MTG = multicomponent training group; ATG = aerobic training group; CVDs = cardiovascular diseases; RHR = resting heart rate; P-SBP = peripheral systolic blood pressure; P-DBP = peripheral diastolic blood pressure; 30 CST = 30-s chair stand test; HGS = hand grip strength; 2MST = 2 min step test; PACES = physical activity enjoyment scale; (D)HGS = dominant hand grip strength; HB = home based; RPE = rate of perceived exertion; IPAQ-SF = International Physical Activity Questionnaire—Short Form; FFP = fried frailty phenotype.

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
