# Peer review of "The Effects of Multicomponent Training on Clinical, Functional, and Psychological Outcomes in Cardiovascular Disease: A Narrative Review"

_medicina, 2025, doi:10.3390/medicina61050822_

Round 1

Reviewer 1 Report

Comments and Suggestions for Authors

The review aimed to critically evaluate the current evidence regarding the effects of multicomponent training (MCT) on physiological and psychological parameters in cardiovascular patients. Although the manuscript is generally well-structured, several modifications are necessary.

  • The study is classified as a systematic review and should include a PROSPERO registration number. Please write systematic review in the title, abstract, and manuscript. Additionally, any required sections for the systematic review, such as the data extraction section, should be incorporated.
  • A section for assessing the risk of bias should be included in both the Methods and Results sections.
  • In Table 1, please add a column to specify the country where each study was conducted.
  • Many paragraphs are excessively long. If a paragraph exceeds 200 to 300 words, it is likely too lengthy for most readers to comprehend easily. Breaking it into smaller sub-paragraphs can significantly enhance readability. Additionally, the introduction was not divided into paragraphs. Please revise the manuscript to ensure it is organized into appropriate paragraphs.
  • Physical activity in Line 45 should be referred to as physical activity (PA), and any subsequent mentions of physical activity in the text should utilize this abbreviation.
  • In line 186, blood pressure (SBP). Line 244 will be as SBP.
  • Please remove the names of the authors from all sections of the manuscript, including citations such as Baptista et al. (2018) [37] and Gonçalvez et al. (2018) [41],…
  • The discussion section is too long; please summarize it.
  • The rationale for the study should be explained more clearly. Please include references to available data and highlight the existing gaps in the literature.
  • It is suggested that all instances of "we" or “our” be replaced with phrases such as “current study," "this study," or "present study".
  • Moderate proofreading is required.
Comments on the Quality of English Language
  • Moderate proofreading is required.

Author Response

Dear reviewer, the point by point reply is attached as pdf file.

Thank you very much.

Reviewer 2 Report

Comments and Suggestions for Authors - The main question addressed by the research is the comparation of multicomponent interventions, integrating different exercises modalities with than single-component strategies in preventing and managing CVDs.  - I consider the article relevant to the field of cardiac rehabilitation. - In the subject area, optimal management  requires an integrated approach combining pharmacological therapy with lifestyle optimization the structured pre and post cardiac rehabilitation  is very important in reducing recurrence risk and enhancing prognosis patients with CVDs. - This narrative review was carried out following the Narrative Review checklist about108 Comprehensive research on Medline In-Process and other Non-Indexed Citations was 109 conducted, using MEDLINE (PubMed), Web of Science (WoS) and Google Scholar articles.  - The tables and figures are clear.  

Author Response

(The authors gave the same response as above.)

Reviewer 3 Report

Comments and Suggestions for Authors

I have reviewed the manuscript entitled 'The Effects of Multicomponent Training on Clinical, Functional, and Psychological Outcomes in Cardiovascular Disease: A Narrative Review '.

The manuscript is well-designed and presented however several issues should be addressed before further evaluation.

First, the role of telemedicine and mHealth systems should be emphasized in the population for the outcomes of cardiovascular diseases. please mention this issue citing 'Telemedicine: Current Concepts and Future Perceptions', 'The effect of 1-year mean step count on the change in the atherosclerotic cardiovascular disease risk calculation in patients with high cardiovascular risk: a sub-study of the LIGHT randomized clinical trial' and 'Digital Health Interventions in Patient Management Following Acute Coronary Syndrome: A Meta-Analysis of the Literature'.

Please explain the adherence to the lifestyle factors and medical treatment adherence more in detail.

In the discussion section, please mention the importance of patients centered approach to the cardiovascular disease which can be provided by the use of AI or ML based systems. Please mention this issue citing 'The Role of Artificial Intelligence in Coronary Artery Disease and Atrial Fibrillation'.

Author Response

(The authors gave the same response as above.)

Round 2

Reviewer 1 Report

Comments and Suggestions for Authors

The authors tried to address the major comments. The manuscript has been revised and shows potential for publication.

Comments on the Quality of English Language

Minor proofreading is required.

Author Response

Dear reviewer,

Thank you very much for your effort. We have done some minor proofreading, as suggest.

In the manuscript, revisions have been highlighted in yellow.

Reviewer 3 Report

Comments and Suggestions for Authors

I have reviewed the manuscript entitled 'The Effects of Multicomponent Training on Clinical, Functional, and Psychological Outcomes in Cardiovascular Disease: A Narrative Review '.

The manuscript is well-designed and presented however several issues should be addressed before further evaluation.

First, the role of telemedicine and mHealth systems should be emphasized in the population for the outcomes of cardiovascular diseases. please mention this issue citing 'Telemedicine: Current Concepts and Future Perceptions', 'The effect of 1-year mean step count on the change in the atherosclerotic cardiovascular disease risk calculation in patients with high cardiovascular risk: a sub-study of the LIGHT randomized clinical trial' and 'Digital Health Interventions in Patient Management Following Acute Coronary Syndrome: A Meta-Analysis of the Literature'.

Please explain the adherence to the lifestyle factors and medical treatment adherence more in detail.

In the discussion section, please mention the importance of patients centered approach to the cardiovascular disease which can be provided by the use of AI or ML based systems. Please mention this issue citing 'The Role of Artificial Intelligence in Coronary Artery Disease and Atrial Fibrillation'.

Author Response

Dear Reviewer,

we have noticed that the comments provided in this second round are identical to those made during the first one. Therefore, it is possible that this may have occurred due to an unintentional error. For convenience, we are re-submitting the responses we had previously provided during the first round:

Point 1: First, the role of telemedicine and mHealth systems should be emphasized in the population for the outcomes of cardiovascular diseases. Please mention this issue citing 'Telemedicine: Current Concepts and Future Perceptions', 'The effect of 1-year mean step count on the change in the atherosclerotic cardiovascular disease risk calculation in patients with high cardiovascular risk: a sub-study of the LIGHT randomized clinical trial' and 'Digital Health Interventions in Patient Management Following Acute Coronary Syndrome: A Meta-Analysis of the Literature'.

Response 1: Thanks for your suggestions. However, we have intentionally excluded all the study that did not performed supervised intervention in presence. Therefore, these references would not match the purpose of our review.

Point 2: Please explain the adherence to the lifestyle factors and medical treatment adherence more in detail.

Response 2: Thank you for the suggestions. However, we have investigated only the adherence to the exercise session, lifestyle and medical treatment adherences were not evaluated.

Point 3: in the discussion section, please mention the importance of patients centered approach to the cardiovascular disease which can be provided by the use of AI or ML based systems. Please mention this issue citing 'The Role of Artificial Intelligence in Coronary Artery Disease and Atrial Fibrillation'.

Response 3: Thank you for the observation, however for the same reasons sated at point 1, we think that these arguments would misleading the reader.

Thank you very much for your effort.